# Comparative Assessment of Antimicrobial Efficacy of Seven Surface Disinfectants against Eight Bacterial Strains in Saudi Arabia: An In Vitro Study

**Naif A. Jalal [1], Rozan A. Al-Atyyani [1], Hamdi M. Al-Said [1,*], Sami S. Ashgar [1], Hani Faidah [1], Ayman K. Johargy [1], Aiman M. Momenah [1], Abeer A. Barhameen [1], Sumyya H. Hariri [1], Farkad Bantun [1], Fadi S. Qashqari [1], Elshiekh B. Khidir [2] and Mohammed H. Althagafi [1]**

[1] Department of Microbiology, Faculty of Medicine, Umm Al-Qura University, Makkah 21955, Saudi Arabia; najalal@uqu.edu.sa (N.A.J.); s44285229@st.uqu.edu.sa (R.A.A.-A.); ssashgar@uqu.edu.sa (S.S.A.); hsfaidah@uqu.edu.sa (H.F.); akjohargy@uqu.edu.sa (A.K.J.); ammomenah@uqu.edu.sa (A.M.M.); aabarhameen@uqu.edu.sa (A.A.B.); shbhariri@uqu.edu.sa (S.H.H.); fmbantun@uqu.edu.sa (F.B.); fsqashqari@uqu.edu.sa (F.S.Q.); mhthagafi@uqu.edu.sa (M.H.A.)

[2] Department of Laboratory Medicine, Faculty of Applied Medical Science, Umm Al-Qura University, Makkah 21955, Saudi Arabia; ebkhidir@uqu.edu.sa

* Correspondence: hmibrahim@uqu.edu.sa; Tel.: +966-551941735

**Abstract:** Environmental conditions in hospitals facilitate the growth and spread of pathogenic bacteria on surfaces such as floors, bed rails, air ventilation units, and mobile elements. These pathogens may be eliminated with proper disinfecting processes, including the use of appropriate surface disinfectants. In this study, we aimed to evaluate of the antibacterial effects of seven surface disinfectants (HAMAYA, DAC, AJAX, Jif, Mr. MUSCLE, CLOROX, and BACTIL) against eight bacterial strains *Klebsiella pneumoniae*, *Enterobacter aerogenes*, *Acinetobacter baumannii*, *Serratia marcescens*, *Escherichia coli*, vancomycin-resistant *Enterococcus faecalis*-ATCC 51299, methicillin-resistant *Staphylococcus aureus*-ATCC 43300, and *Pseudomonas aeruginosa*-ATCC 1544, using two methods. The first was to determine the effective contact time of disinfectant against the tested bacterial strains, and the second was an assessment of the disinfection efficacy of each disinfectant on six types of contaminated surfaces with on a mixture of the eight tested bacterial strains. The results showed the efficacy of the disinfectants against the tested strains depending on the effective contact time. BACTIL disinfectant showed an efficacy of 100% against all tested strains at the end of the first minute of contact time. HAMAYA, DAC, Jif, Mr. MUSCLE, and CLOROX showed 100% efficiency at the end of the fourth, fifth, sixth, seventh, and fourteenth minutes, respectively, while AJAX disinfectant required nineteen minutes of contact time to show 100% efficacy against all tested strains.

**Keywords:** antimicrobial efficacy; surface disinfectants; disinfection efficiency

## 1. Introduction

In recent years, hospital-acquired infections have become an increasing cause of morbidity and mortality worldwide [1]. More than 1.7 million annual infections and 100,000 deaths in the United States are due to microbial infections in healthcare settings [2]. According to a report published by the European Centre for Disease Prevention and Control (ECDC), there are 33,000 deaths every year in the European Union due to multidrug-resistant bacterial infections [3]. Public health has been impacted by serious threats from microbial strains present in hospital environments, especially multidrug-resistant (MDR) pathogenic bacterial species such as *Pseudomonas aeruginosa*, *Acinetobacter baumannii*, some Enterobacteriaceae strains, vancomycin-resistant *Enterococcus* spp. (VRE), methicillin-resistant *Staphylococcus aureus* (MRSA), and *Clostridium difficile* [4].

Environmental conditions in hospitals, as well as the ability of pathogens to adhere to surfaces and form biofilms within 24 h, contribute to the spread of pathogens through

contact with medical devices and other surfaces, such as floors and bed rails, air ventilation units, and mobile elements [5–7]. With the emergence of antibiotic-resistant bacterial infections, the use of disinfectants in hospitals and healthcare settings has increased to reduce the pathogens to safe levels and to minimize the transmission of infectious diseases [8].

Disinfectants are antimicrobial products used to kill harmful microorganisms and contain one or more active substances, such as quaternary ammonium, alcohols, sodium hypochlorite, hydrogen peroxide, triclosan phenol, and aldehydes [9,10]. Several medical reports have described an increase in the prevalence of multidrug-resistant organisms (MDROs) during the COVID-19 pandemic [11–14]. A recent Jordanian study reported an increasing prevalence of multidrug- resistant bacterial infections in hospitals in Saudi Arabia, Jordan, Oman, Iran, and Qatar, associated with higher rates of morbidity and mortality, the most common of which were surgical site infections (SSIs) (50%) and bloodstream infections (BSIs) (50%) [15]. Another study conducted in four hospitals in the Hail region of Saudi Arabia, published in 2021, showed that the prevalence of multidrug resistance in Gram-negative bacteria in intensive care units was 30% [16].

A study by the Saudi Food and Drug Authority (SFDA) on the efficacy of chlorohexidine disinfectant showed effectiveness against Gram-negative bacteria [17]. An Egyptian study suggested the need to avoid suboptimal decontamination practices, such as insufficient exposure times and concentrations [18]. Some studies have reported that proper disinfection is linked to many factors, such as disinfectant contact time, correct dilution, nonuse of the same disinfectant at the same dose for long periods to avoid the emergence of resistance, and choosing the right product [19]. A recently-published study showed that molecular biology could be used as one method for effective airway disinfection by decomposing deoxyribonucleic acid (DNA) into smaller fragments in Gram-negative bacteria [20].

Infection control in hospitals requires highly effective disinfection procedures since the spread and persistence of infection is partially due to the usage of incorrect methods of disinfection [21–23]. Consequently, this study aims to evaluate of the antimicrobial efficacy of seven types of disinfectant against eight pathogenic bacterial strains collected in Makkah, Saudi Arabia.

## 2. Materials and Methods

### 2.1. Study Design

This study was conducted at the Microbiology Research Center of the College of Medicine, Umm Al-Qura University, Makkah, Saudi Arabia, during the period from January 2022 to June 2022, to evaluate the antibacterial effect of seven types of disinfectants against eight pathogenic bacterial strains, using two methods. The first was to determine the effective contact time of the disinfectants for the tested bacterial strains, and the second was to assess the disinfection efficacy depending on the effective contact times of each disinfectant on six types of surface contaminated with the tested bacterial strains.

### 2.2. Tested Bacterial Strains

The most important bacterial strains that cause nosocomial infections in Saudi Arabia were include five multidrug-resistant bacterial strains (clinical isolates): *Klebsiella pneumoniae*, *Enterobacter aerogenes*, *Acinetobacter baumannii*, *Serratia marcescens*, and *Escherichia coli.* These strains were collected from tertiary-care hospitals in Makkah, and three reference strains were obtained from the Microbiology Research Center, Faculty of Medicine, Umm Al-Qura University, Makkah: vancomycin-resistant *Enterococcus faecalis* (VRE) ATCC 51299, methicillin-resistant *Staphylococcus aureus* (MRSA) ATCC 43300, and *Pseudomonas aeruginosa* ATCC 15442.

## 2.3. Disinfectants

Seven disinfectants from three classes of phenolic compounds (BACTIL®), quaternary ammonium compounds (HAMAYA®, DAC®, AJAX®, and Jif®), and sodium hypochlorite compounds (Mr. MUSCLE® and CLOROX® disinfectant) were used in the current study at ready-to-use concentrations for spraying and wiping on surfaces. These are the most common products available in Saudi Arabian markets for disinfection in hospitals and health centers, as described in Table 1.

**Table 1.** Characteristics of the tested disinfectants.

| Trade Name | Manufacturer's Description * | Category and Ingredients | Advantages | Disadvantages |
|---|---|---|---|---|
| BACTIL® | General disinfectant liquid, with a refreshing fragrance, that can eliminate virulent and multidrug-resistant strains. | Phenolic compounds [Essential oil components less than 5%] | -Good efficacy with organic material. -Effective over a large pH range. -Stable in storage. -Bactericidal and fungicidal action. | -Causes skin and eye irritation. -Unpleasant odor. -Effectiveness reduced by alkaline pH. |
| HAMAYA® | Used for rapid disinfection of all hard surfaces. | Quaternary ammonium compounds (QACs). | -Non-irritating to skin. -Effective at high temperatures and high pH levels. -Stable in storage. -Rapid action. -Colorless, odorless, non-toxic, highly stable. | -Not effective on non-enveloped viruses, TB bacteria, and spores. -Effectiveness influenced by water hardness. |
| DAC® | Disinfectant and cleaning product, its formula is developed to kill 99.9% of germs and bacteria. | QACs. [Alkyl benzyl dimethyl ammonium chloride < 5% Didecydimethyl ammonium chloride (8.5%)] | | |
| AJAX® | Kills 99.9% of bacteria, can remove grease and limescale for shiny clean surfaces, and leaves a pleasant fragrance. | QACs. | | |
| Jif® | Kills 99.9% of bacteria and germs. | QACs (Benzalkonium chloride) | | |
| Mr. MUSCLE® | Can be used on all surfaces in bathrooms, living rooms, bedrooms; on walls, floors, and shelves. | Sodium hypochlorite (1.1% active chlorine). | -Broad spectrum. -Inexpensive. -Penetrates the cell wall quickly. -Kills a wide range of microorganisms. | -Inactivated by sunlight, some metals. -Irritating to mucous membranes, skin. -Tuberculocidal with extended contact time. -Organics may reduce activity. -An increase in alkalinity decreases bactericidal properties. |
| CLOROX® | Can kill 99.9% of germs. | Sodium hypochlorite (6.0%). | | |

Note: * The information in this column is as mentioned on the disinfectant labels.

## 2.4. Culture Media

Mueller-Hinton agar (MHA) and Mueller-Hinton broth (MHB) were used in the current study; MHB was used to perform the contact-time assay and assessment of the efficacy of the disinfectants against the bacterial strains on six types of contaminated surfaces. MHA was used as a culture medium for determination of the efficacy of the disinfectants against the tested bacterial strains during a given contact time.

## 2.5. Preparation of Bacterial Suspension

The bacterial strain was cultured in Mueller-Hinton broth and incubated at 37 °C for 24 h. A loopful of bacterial suspension was transferred in a tube containing 3 mL broth and adjusted to a turbidity equivalent of 0.5 McFarland Standard, using a calibrated VITEK 2 DENSICHEK, according to Clinical Laboratory Standards Institute (CLSI) guidelines [24].

*2.6. Determination of the Contact Time of Each Disinfectant against Tested Bacterial Strains*

Using a sterile 96-well microtiter plate, 200 µL of the tested disinfectant was added to the first and second wells in the first column of the microtiter plate, and 100 µL of MH broth was added to the other wells. Next, 20 µL of the tested bacterial suspension in MH broth, adjusted to 0.5 McFarland, was pipetted for each strain in the disinfectant wells (first and second wells) in the first column. After 30 s, 10 µL was transferred from the well that contained the disinfectant to the next well, containing 100 µL of Mueller-Hinton broth.

This step was repeated every 30 s (from the well-containing disinfectant to new wells containing 100 µL of MHB), a total of 40 times, for the 40 wells containing 100 µL of MHB, over time periods ranging from 30 s to 20 min; Mueller-Hinton broth alone was used as the negative control, and 10 µL of the tested bacterial suspension in only Mueller-Hinton broth was used as positive control. This process was repeated for each disinfectant against each tested bacterial strain.

The microtiter plates were incubated at 37 °C overnight. After the incubation period, 10 µL from each well was transferred to a subculture on a Mueller-Hinton agar plate, and incubated at 37 °C overnight.

On the third day, the agar plates were examined for bacterial growth. An effective disinfectant was defined by an absence of growth on the agar plate. The presence of bacterial growth indicated that there was no effect of the disinfectant against the tested bacterial strain at the recorded time. This experiment was repeated in two replicates on different days.

*2.7. Assessment of the Disinfection Efficacy of Each Disinfectant on Six Types of Contaminated Surfaces with a Mixture of Eight Bacterial Strains*

The surfaces used in this study were wood, glass, leather, plastic, marble, and stainless steel. They were divided into equal square areas (15 × 15 cm). A mixture of the eight tested bacterial strains used in the previous contact time assay was prepared in Mueller-Hinton broth and adjusted to 0.5 McFarland standard via automated McFarland measurement. A total of 200 µL of the bacterial mixture was transferred and sprayed on every surface and left to dry. Then, 200 µL of each disinfectant was sprayed on each of the six different surfaces contaminated with the bacterial mixture and left for periods ranging from 30 s to 20 min, according to the previous results of the effective contact time assay for every disinfectant.

Three swabs (sterile cotton swabs) were collected from each contaminated surface, two minutes before the effective contact time, two minutes after the end of the effective contact time, and at the effective contact time, except for the surfaces that were sprayed with the BACTIL disinfectant (because the effective contact time of BACTIL was one minute against tested bacterial strains), from which swabs were collected at 30 s before the effective contact time, at the effective contact time of 60 s, and 30 s after the end of the effective contact time. These swabs were placed in three tubes, each containing 2 mL Mueller-Hinton broth, and mixed well. Then, 100 µL of each tube was transferred for culturing on a Mueller-Hinton agar plate and incubated at 37 °C for 24 h to determine the bacterial growth.

This step was repeated for each disinfectant against each contaminated surface. The experiment was repeated in two replicates on different days during the study period.

## 3. Results

*3.1. Effective Contact Time of Disinfectants against Tested Bacterial Strains*

In the current study we determined the antibacterial activity of seven disinfectants (HAMAYA, DAC, AJAX, Jif, Mr. MUSCLE, CLOROX, and BACTIL) against five bacterial strains (clinical isolates, *Klebsiella pneumoniae*, *Enterobacter aerogenes*, *Acinetobacter baumannii*, *Serratia marcescens*, and *Escherichia coli*), and three bacterial strains (reference strains, vancomycin-resistant *Enterococcus faecalis*-ATCC 51299, methicillin-resistant *Staphylococcus aureus*-ATCC 43300, and *Pseudomonas aeruginosa*-ATCC 1544), through time periods ranging from 30 s to 19 min, as shown in Tables 2 and 3.

**Table 2.** Evaluation of the antibacterial activity of seven disinfectants against five tested bacterial strains (clinical isolates) by contact-time assay.

| Tested Strains | Disinfectants | Contact Time (min ± SD) | Effective Contact Time (min) |
|---|---|---|---|
| *Enterobacter aerogenes* | HAMAYA | 2 ± 0.50 | 3 |
| | DAC | 2 ± 0.00 | 2 |
| | AJAX | 15 ± 1.00 | 16 |
| | Jif | 5 ± 0.50 | 6 |
| | Mr. MUSCLE | 4 ± 1.00 | 5 |
| | CLOROX | 10 ± 0.50 | 11 |
| | BACTIL | 0.5 ± 0.00 | 30 s |
| *Escherichia coli* | HAMAYA | 2 ± 0.50 | 3 |
| | DAC | 3 ± 0.00 | 3 |
| | AJAX | 15 ± 1.00 | 16 |
| | Jif | 5 ± 0.00 | 5 |
| | Mr. MUSCLE | 5 ± 0.50 | 6 |
| | CLOROX | 10 ± 1.50 | 12 |
| | BACTIL | 0.5 ± 0.00 | 30 s |
| *Klebsiella pneumoniae* | HAMAYA | 3 ± 0.50 | 4 |
| | DAC | 3 ± 1.00 | 4 |
| | AJAX | 15 ± 0.50 | 16 |
| | Jif | 5 ± 0.50 | 6 |
| | Mr. MUSCLE | 5 ± 1.00 | 6 |
| | CLOROX | 10 ± 1.50 | 12 |
| | BACTIL | 0.5 ± 0.00 | 30 s |
| *Acinetobacter baumannii* | HAMAYA | 1.5 ± 0.50 | 2 |
| | DAC | 2 ± 0.00 | 2 |
| | AJAX | 15 ± 1.00 | 16 |
| | Jif | 5 ± 0.00 | 5 |
| | Mr. MUSCLE | 5 ± 1.00 | 6 |
| | CLOROX | 10 ± 0.50 | 11 |
| | BACTIL | 0.5 ± 0.00 | 30 s |
| *Serratia marcescens* | HAMAYA | 2 ± 0.50 | 3 |
| | DAC | 2 ± 0.00 | 2 |
| | AJAX | 15 ± 1.00 | 16 |
| | Jif | 5 ± 0.00 | 5 |
| | Mr. MUSCLE | 5 ± 1.00 | 6 |
| | CLOROX | 10 ± 0.50 | 11 |
| | BACTIL | 0.5 ± 0.00 | 30 s |

**Table 3.** Evaluation of the antibacterial activity of seven disinfectants against three tested bacterial strains (ATCC reference strains) by contact time assay.

| Tested Strains | Disinfectants | Contact Time (min ± SD) | Effective Contact Time (min) |
|---|---|---|---|
| Methicillin-resistant *Staphylococcus aureus* ATCC 43300 | HAMAYA | 2 ± 1.00 | 3 |
| | DAC | 2 ± 0.50 | 3 |
| | AJAX | 14 ± 1.00 | 15 |
| | Jif | 5 ± 0.50 | 6 |
| | Mr. MUSCLE | 4 ± 0.50 | 5 |
| | CLOROX | 9 ± 0.50 | 10 |
| | BACTIL | 0.5 ± 0.00 | 30 s |
| Vancomycin-resistant *Enterococcus faecalis* ATCC 51299 | HAMAYA | 1 ± 0.50 | 2 |
| | DAC | 2 ± 1.00 | 3 |
| | AJAX | 13 ± 2.00 | 15 |
| | Jif | 4 ± 1.00 | 5 |
| | Mr. MUSCLE | 4 ± 1.00 | 5 |
| | CLOROX | 8 ± 1.50 | 10 |
| | BACTIL | 0.5 ± 0.00 | 30 s |
| *Pseudomonas aeruginosa* ATCC 1544 | HAMAYA | 3 ± 1.00 | 4 |
| | DAC | 4 ± 0.50 | 5 |
| | AJAX | 17 ± 2.00 | 19 |
| | Jif | 5 ± 1.00 | 6 |
| | Mr. MUSCLE | 6 ± 0.50 | 7 |
| | CLOROX | 12 ± 1.50 | 14 |
| | BACTIL | 1 ± 0.00 | 1 |

HAMAYA disinfectant showed efficacy against *Enterobacter aerogenes*, *Escherichia coli*, *Klebsiella pneumoniae*, *Acinetobacter baumannii*, *Serratia marcescens*, methicillin-resistant *Staphylococcus aureus* ATCC 43300, vancomycin-resistant *Enterococcus faecalis* ATCC 51299, and *Pseudomonas aeruginosa* ATCC 1544 at 2 ± 0.50, 2 ± 0.50, 3 ± 0.50, 1.5 ± 0.50, 2 ± 0.50, 2 ± 1.00, 1 ± 0.50, and 3 ± 1.00 min, respectively of contact time.

DAC disinfectant showed efficacy against *Enterobacter aerogenes*, *Escherichia coli*, *Klebsiella pneumoniae*, *Acinetobacter baumannii*, *Serratia marcescens*, methicillin-resistant *Staphylococcus aureus* ATCC 43300, vancomycin-resistant *Enterococcus faecalis* ATCC 51299, and *Pseudomonas aeruginosa* ATCC 1544 at 2 ± 0.00, 3 ± 0.00, 3 ± 1.00, 3 ± 0.50, 2 ± 0.00, 2 ± 0.50, 2 ± 1.00, and 4 ± 0.50 min, respectively.

AJAX disinfectant showed efficacy against *Enterobacter aerogenes*, *Escherichia coli*, *Klebsiella pneumoniae*, *Acinetobacter baumannii*, *Serratia marcescens*, methicillin-resistant *Staphylococcus aureus* ATCC 43300, vancomycin-resistant *Enterococcus faecalis* ATCC 51299, and *Pseudomonas aeruginosa* ATCC 1544 at 15 ± 1.00, 15 ± 1.00, 15 ± 0.50, 15 ± 0.50, 15 ± 1.00, 14 ± 1.00, 13 ± 2.00, and 17 ± 2.00 min, respectively.

Jif disinfectant exhibited effectiveness at 5 ± 0.50, 5 ± 0.00, 5 ± 0.50, 5 ± 1.00, 5 ± 0.00, 5 ± 0.50, 4 ± 1.00, and 5 ± 1.00 min against *Enterobacter aerogenes*, *Escherichia coli*, *Klebsiella pneumoniae*, *Acinetobacter baumannii*, *Serratia marcescens*, methicillin-resistant *Staphylococcus aureus* ATCC 43300, vancomycin-resistant *Enterococcus faecalis* ATCC 51299, and *Pseudomonas aeruginosa* ATCC 1544, respectively.

Mr. MUSCLE disinfectant showed efficacy at $4 \pm 1.00$, $5 \pm 0.50$, $5 \pm 1.00$, $5 \pm 0.50$, $5 \pm 1.00$, $4 \pm 0.50$, $4 \pm 1.00$, and $6 \pm 0.50$ min of contact time against *Enterobacter aerogenes*, *Escherichia coli*, *Klebsiella pneumoniae*, *Acinetobacter baumannii*, *Serratia marcescens*, methicillin-resistant *Staphylococcus aureus* ATCC 43300, vancomycin-resistant *Enterococcus faecalis* ATCC 51299, and *Pseudomonas aeruginosa* ATCC 1544, respectively.

CLOROX disinfectant was effective against both vancomycin-resistant *Enterococcus faecalis* ATCC 51299 and methicillin-resistant *Staphylococcus aureus* ATCC 43300 at $8 \pm 1.50$ and $9 \pm 0.50$ min, respectively, at $10 \pm 0.50$ min against *Serratia marcescens*, *Acinetobacter baumannii*, and *Enterobacter aerogenes*, at $10 \pm 1.50$ against *Klebsiella pneumoniae and Escherichia coli*, and at $12 \pm 1.50$ against *Pseudomonas aeruginosa* ATCC 1544.

BACTIL disinfectant showed efficacy against *Enterobacter aerogenes*, *Escherichia coli*, *Klebsiella pneumoniae*, *Acinetobacter baumannii*, *Serratia marcescens*, methicillin-resistant *Staphylococcus aureus* ATCC 43300, vancomycin-resistant *Enterococcus faecalis* ATCC 51299, and *Pseudomonas aeruginosa* ATCC 1544 at $0.5 \pm 0.00$, $0.5 \pm 0.00$, $0.5 \pm 0.00$, $0.5 \pm 0.00$, $0.5 \pm 0.00$, $0.5 \pm 0.00$, $0.5 \pm 0.00$, and $1 \pm 0.00$ min, respectively (Tables 2–4).

**Table 4.** Determination of effective contact time of the seven disinfectants against eight bacterial strains.

| Tested Strains | Quaternary Ammonium Compounds | | | | Sodium Hypochlorite | | Phenolic Compounds |
|---|---|---|---|---|---|---|---|
| | HAMAYA | DAC | Jif | AJAX | Mr. MUSCLE | CLOROX | BACTIL |
| *E. aerogenes* | 3 min | 2 min | 16 min | 6 min | 5 min | 11 min | 30 s |
| *E. coli* | 3 min | 3 min | 16 min | 5 min | 6 min | 12 min | 30 s |
| *K. pneumoniae* | 4 min | 4 min | 16 min | 6 min | 6 min | 12 min | 30 s |
| *A. baumannii* | 2 min | 2 min | 16 min | 5 min | 6 min | 11 min | 30 s |
| *S. marcescens* | 3 min | 2 min | 16 min | 5 min | 6 min | 11 min | 30 s |
| *P. aeruginosa*-ATCC 1544 | 4 min | 5 min | 19 min | 6 min | 7 min | 14 min | 1 min |
| MRSA-ATCC 43300 | 3 min | 3 min | 15 min | 6 min | 5 min | 10 min | 30 s |
| VRE-51299 | 2 min | 3 min | 15 min | 5 min | 5 min | 10 min | 30 s |
| Effective Contact Time | 4 min | 5 min | 19 min | 6 min | 7 min | 14 min | 1 min |

*3.2. Determination of Quantitative Values of Disinfectant Efficacy against Eight Bacterial Strains within 20 Min of Contact Time*

BACTIL had the highest efficacy (100%) against all tested bacterial strains at the first minute. In the second and third minutes, HAMAYA and DAC showed efficacies of 30% and 62.5%, respectively, and at the fourth minute showed efficacies of 100% and 75%, respectively.

In the fifth minute, DAC, Jif, and Mr. MUSCLE showed efficacies of 100%, 37.5%, and 37.5%, respectively, against the eight tested strains.

In the sixth minute, the efficacy of Jif and Mr. MUSCLE increased to 100% and 87.5%, respectively. In the seventh minute, the efficacy of Mr. MUSCLE increased to 100%.

In the 10th, 11th, 12th, 13th, and 14th minutes of contact time, the efficacy of CLOROX increased from 25% to 62.5%, 87.5%, and 87.5% to 100%.

In the 15th, 16th, 17th, 18th, and 19th minutes of contact time, the efficacy of AJAX increased from 25% to 87.5%, 87.5%, 87.5%, and 100% (Table 5).

**Table 5.** Determination of quantitative values of the disinfection efficiency of seven disinfectants against eight tested bacterial strains within 20 min.

| Contact Time/min | Efficacy Rate (%) | | | | | | |
|---|---|---|---|---|---|---|---|
| | HAMAYA | DAC | Jif | AJAX | Mr. MUSCLE | CLOROX | BACTIL |
| 1 | 0 | 0 | 0 | 0 | 0 | 0 | 100% |
| 2 | 30% | 30% | 0 | 0 | 0 | 0 | 100% |
| 3 | 62.5% | 62.5% | 0 | 0 | 0 | 0 | 100% |
| 4 | 100% | 75% | 0 | 0 | 0 | 0 | 100% |
| 5 | 100% | 100% | 37.5% | 0 | 37.5% | 0 | 100% |
| 6 | 100% | 100% | 100% | 0 | 87.5% | 0 | 100% |
| 7 | 100% | 100% | 100% | 0 | 100% | 0 | 100% |
| 8 | 100% | 100% | 100% | 0 | 100% | 0 | 100% |
| 9 | 100% | 100% | 100% | 0 | 100% | 0 | 100% |
| 10 | 100% | 100% | 100% | 0 | 100% | 25% | 100% |
| 11 | 100% | 100% | 100% | 0 | 100% | 62.5% | 100% |
| 12 | 100% | 100% | 100% | 0 | 100% | 87.50% | 100% |
| 13 | 100% | 100% | 100% | 0 | 100% | 87.50% | 100% |
| 14 | 100% | 100% | 100% | 0 | 100% | 100% | 100% |
| 15 | 100% | 100% | 100% | 25% | 100% | 100% | 100% |
| 16 | 100% | 100% | 100% | 87.5% | 100% | 100% | 100% |
| 17 | 100% | 100% | 100% | 87.5% | 100% | 100% | 100% |
| 18 | 100% | 100% | 100% | 87.5% | 100% | 100% | 100% |
| 19 | 100% | 100% | 100% | 100% | 100% | 100% | 100% |
| 20 | 100% | 100% | 100% | 100% | 100% | 100% | 100% |

*3.3. Assessment of the Disinfection Effectiveness of the Seven Disinfectants on Six Types of Contaminated Surfaces with a Mixture of Eight Bacterial Strains*

The results show no difference between the effective contact time of each disinfectant on all contaminated surfaces and the contact time recorded in the previous test (contact time) for the same disinfectant against the tested strains (Table 6).

**Table 6.** Disinfection efficacy of each disinfectant against six contaminated surfaces with eight bacterial strains.

| Disinfectants | Contact Time/min | Type of Contaminated Surface | | | | | |
|---|---|---|---|---|---|---|---|
| | | Glass | Wood | Marble | Plastic | Leather | Stainless Steel |
| HAMAYA | 2 min | Not Effective | Not Effective | Not Effective | Not Effective | Not Effective | Not Effective |
| | 4 min | Effective | Effective | Effective | Effective | Effective | Effective |
| | 6 min | Effective | Effective | Effective | Effective | Effective | Effective |
| DAC | 3 min | Not Effective | Not Effective | Not Effective | Not Effective | Not Effective | Not Effective |
| | 5 min | Effective | Effective | Effective | Effective | Effective | Effective |
| | 7 min | Effective | Effective | Effective | Effective | Effective | Effective |
| Jif | 4 min | Not Effective | Not Effective | Not Effective | Not Effective | Not Effective | Not Effective |
| | 6 min | Effective | Effective | Effective | Effective | Effective | Effective |
| | 8 min | Effective | Effective | Effective | Effective | Effective | Effective |

**Table 6.** *Cont.*

| Disinfectants | Contact Time/min | Type of Contaminated Surface | | | | | |
|---|---|---|---|---|---|---|---|
| | | Glass | Wood | Marble | Plastic | Leather | Stainless Steel |
| AJAX | 17 min | Not Effective | Not Effective | Not Effective | Not Effective | Not Effective | Not Effective |
| | 19 min | Effective | Effective | Effective | Effective | Effective | Effective |
| | 21 min | Effective | Effective | Effective | Effective | Effective | Effective |
| Mr. MUSCLE | 5 min | Not Effective | Not Effective | Not Effective | Not Effective | Not Effective | Not Effective |
| | 7 min | Effective | Effective | Effective | Effective | Effective | Effective |
| | 9 min | Effective | Effective | Effective | Effective | Effective | Effective |
| CLOROX | 12 min | Not Effective | Not Effective | Not Effective | Not Effective | Not Effective | Not Effective |
| | 14 min | Effective | Effective | Effective | Effective | Effective | Effective |
| | 16 min | Effective | Effective | Effective | Effective | Effective | Effective |
| BACTIL | 30 s | Not Effective | Not Effective | Not Effective | Not Effective | Not Effective | Not Effective |
| | 60 s | Effective | Effective | Effective | Effective | Effective | Effective |
| | 90 s | Effective | Effective | Effective | Effective | Effective | Effective |

Note: Not Effective = bacterial growth. Effective = no bacterial growth.

## 4. Discussion

Disinfectants in hospitals are an important tool for combating the spread of infectious diseases when used correctly and in accordance with instructions [25–28]. Proper disinfection protocols in hospitals and healthcare environments are essential for minimizing the risks of infection, especially given the increasing prevalence of MDR organisms [29]. Therefore, surveillance of the rising prevalence of multidrug- resistant bacterial infections in hospitals is essential worldwide in order to take early measures to limit their spread and impact [30]. Proper disinfection depends on several factors such as the contact time (exposure time) of the disinfectant on pathogens, potency, and the concentration of the disinfectant [4,23]. Due to the limited studies in Saudi Arabia on the contact time factor, we aimed to evaluate the antibacterial efficacy of seven disinfectants against eight MDR bacterial strains via contact testing.

The results showed the efficacy of the disinfectants against the tested bacterial strains according to the effective contact time for each disinfectant. For example, BACTIL (phenolic-based) showed an efficacy of 100% against all tested strains at the end of the first minute of contact time. HAMAYA, DAC, Jif, and AJAX (QAC-based) showed 100% efficiency at the end of the fourth, fifth, sixth, and nineteenth minutes, respectively, while Mr. MUSCLE and CLOROX (sodium-hypochlorite-based) required seven and fourteen minutes, respectively, contact time to show 100% efficacy against all the tested strains.

Therefore, the current results confirm that Gram-negative bacteria were more resistant to the seven disinfectants than Gram-positive bacteria, especially *P. aeruginosa* ATCC 1544, which required a contact time with HAMAYA, DAC, Jif, and AJAX of 4, 5, 6, and 19 min, respectively. Mr. MUSCLE and CLOROX required 7 and 14 min, respectively, and BACTIL required one minute to eliminate all Gram-negative bacteria. However, the Gram-positive bacterial strains methicillin-resistant *Staphylococcus aureus* ATCC 43300 and vancomycin-resistant *Enterococcus faecalis* ATCC 51299 required contact times of fifteen minutes, ten minutes and 30 s for quaternary ammonium compounds, sodium hypochlorite, and phenolic compounds, to eliminate them.

These variations in the effective contact time can be attributed to the chemical and physical characteristics of the microbial cell surface, the genus of the bacterial strain, the concentration of the disinfectant, its mechanism of action, and the mechanism of bacterial resistance [23,31,32]. Furthermore, the Center for Disease Control and Prevention's (CDC) guidelines for disinfection in healthcare facilities identify many factors that may affect disinfection efficacy such as contact time [33]. In this study, the contact time was higher than those reported in a study from Saudi Arabia in 2020 [34], on the efficacy of a number

of disinfectants (QACs and sodium hypochlorite) against thirteen bacterial strains (Gram-positive and negative). That study showed that the tested bacterial strains were sensitive to DAC and CLOROX disinfectant at the first minute of the contact time, while *Klebsiella pneumoniae* responded to CLOROX after the five minute contact time.

Our study shows that disinfectants based on QACs were effective against six strains of Gram-negative bacteria following between 2 and 19 min of contact time and against two Gram-positive strains following between 2 and 15 min of contact time. DAC was effective against six strains of Gram-negative bacteria (*E. aerogenes*, *E. coli*, *K. pneumoniae*, *A. baumannii*, *S. marcescens*, and *P. aeruginosa*) at 2, 3, 4, 2, 2, and 5 min, respectively, and two strains of Gram-positive bacteria (MRSA-ATCC 43300 and VR E-ATCC 51299) at 3 and 2 min. Disinfectants based on sodium hypochlorite were effective against six strains of Gram-negative bacteria between 5 and 14 min and two Gram-positive strains between 5 and 10 min of contact time. CLOROX showed efficacy against six strains of Gram-negative bacteria (*E. aerogenes*, *E. coli*, *K. pneumoniae*, *A baumannii*, *S. marcescens*, and *P. aeruginosa*) at 11, 12, 12, 11, 11, and 14 min, respectively, and two strains of Gram-positive bacteria (MRSA-ATCC 43300 and VR E-ATCC 51299) at 10 min for both. Disinfectants based on phenolic compounds (BACTIL) were effective against six strains of Gram-negative bacteria at the end of the first minute and two Gram-positive strains at 30 s of contact time.

This finding was in agreement with previous studies in Saudi Arabia, where it was found that disinfectants based on QACs and sodium hypochlorite were less effective against Gram-negative bacteria than Gram-positive bacteria, especially *P. aeruginosa* [15–17,35]. The same finding was reported in other countries [36–38], including Morocco, Italy, and the USA, which showed that quaternary ammonium disinfectants were more effective against Gram-positive than against Gram-negative bacterial strains, notably *P. aeruginosa*. This is because Gram-negative bacteria have complex cell membranes that are selectively permeable and act as barriers preventing the absorption of antibacterial agents, leading to an increase in their antibiotic-resistance rates. The Gram-negative bacteria's ability to alter their outer membrane through mutations or changes in hydrophobic properties makes them less sensitive to disinfectants than Gram-positive bacteria, which do not possess this ability [39,40].

A previous study showed that the required contact time between the disinfectant and microorganisms should be within a few minutes, due to the speed of evaporation of some disinfectants which leads to the microorganisms not being affected by the disinfectant and building resistance against it [32,33]. Therefore, disinfectants that showed efficacy within a short time period such as BACTIL, HAMAYA, DAC, Jif, and Mr. MUSCLE are more suitable for proper disinfection. These required contact times ranging from 60 s to seven minutes.

## 5. Conclusions

Proper disinfection protocols in hospitals are essential for minimizing the risks of infection, especially given the increasing prevalence of MDR organisms. Proper disinfection depends on several factors such as the contact time (exposure time) of the disinfectant against pathogens. From this perspective, our study focused on determining the effective contact time of seven disinfectants against eight bacterial strains during periods ranging from 30 s to 19 min. The results revealed that Gram-negative bacteria were more resistant to the seven disinfectants than Gram-positive bacteria, especially *P. aeruginosa* ATCC 1544. In addition, BACTIL disinfectant achieved the highest efficacy against the tested strains within 60 s, followed by HAMAYA, DAC, Jif, and Mr. MUSCLE that were showed antibacterial efficacy at the end of the fourth, fifth, sixth, and seventh minutes, respectively. In contrast, Clorox and AJAX disinfectants respectively required fourteen and nineteen minutes of contact time to achieve effectiveness against the tested strains. Moreover, the results showed that the type of surface to be disinfected did not affect the efficiency of the disinfectant used, especially on the types of surfaces that were tested in this study. The results of this study compared with results about the same products from several nearby and distant countries

showed an increase in the rates of emergence and spread of multidrug-resistant pathogens in varying proportions. This may be attributed to many factors including a failure to apply disinfection protocols appropriately. Therefore, the results of the current study could be a reference for disinfection protocols in hospitals and healthcare centers to reduce the spread of MDR pathogens. Based on what was achieved, these results suggest further studies to evaluate other factors that affect the antimicrobial efficacy of surface disinfectants against MDR pathogens, especially since this study was limited to eight bacterial strains. This warrants an increase in the number of tested strains in future studies.

**Author Contributions:** Conceptualization, H.M.A.-S., N.A.J., S.S.A., R.A.A.-A., A.K.J., A.A.B., S.H.H., F.B., H.F. and E.B.K.; Formal analysis, R.A.A.-A.; Funding acquisition, A.M.M.; Investigation, H.M.A.-S., N.A.J., S.S.A., A.K.J., A.M.M., A.A.B., S.H.H., F.B. and H.F.; Methodology, N.A.J., R.A.A.-A., H.M.A.-S., F.S.Q. and M.H.A.; Project administration, H.M.A.-S., N.A.J. and S.H.H.; Resources, H.M.A.-S., N.A.J., A.M.M., E.B.K. and M.H.A.; Supervision, H.M.A.-S., F.S.Q., N.A.J., A.K.J. and H.F.; Visualization, E.B.K.; Writing—original draft, N.A.J., S.S.A., R.A.A.-A., H.M.A.-S. and A.A.B.; Writing—review & editing, H.M.A.-S., N.A.J., F.B. and A.M.M. All authors have read and agreed to the published version of the manuscript.

**Funding:** No funding for this research from any sector.

**Institutional Review Board Statement:** Not applicable.

**Informed Consent Statement:** Not applicable.

**Data Availability Statement:** Not applicable.

**Conflicts of Interest:** The authors declare no conflict of interest.

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
