# Peer review of "Comparative Assessment of Antimicrobial Efficacy of Seven Surface Disinfectants against Eight Bacterial Strains in Saudi Arabia: An In Vitro Study"

_2036-7481, doi:10.3390/microbiolres14030058_

Round 1

Reviewer 1 Report

The article is generally well-written, with some typos highlighted in red (see the attached pdf).

Some comments and suggestions are also reported in the pdf file.

Additionally, the reviewer found some major criticisms:

Since concentration is a pivotal parament in the fight against MDR microbial, how much is the concentration of the microbe in the well? Similarly, how much is the concentration of the antimicrobial agents in the well? This is a crucial point.

How the authors can justify the difference between the results shown in the article in question and those reported in reference 34?

The authors should modify the “Contact Time/Min” parameter since the significant numbers of digits of the values and SD are often unequal (1 vs 3).

From line 306 to line 322: is this part necessary? The reviewer found this part inappropriate in the "Discussion" section. Maybe the authors can move it to another section.

English is good.

Some typos were detected.

Some sentences need to be revised.

Reviewer 2 Report

In the title, in vitro must be in italics.

Moderate editing of English language required.

The products certainly passed tests with specific microorganisms, carried out by the manufacturer. I think it would be good to include them in Table 1, if the information is available.

When 10 uL of the disinfectant + bacteria solution were inoculated in 100 uL of sterile MHB, can the residual concentration of the disinfectants change the survival of the microorganisms? Has this effect been measured?

Scientific names should always be italicized. Correct throughout the manuscript.

What is the contribution to the field of knowledge made by this article? In the discussion, it is worth mentioning again the tests carried out by the manufacturers, in order to show whether the results obtained were expected or not.

Moderate editing of English language required.

Reviewer 3 Report

Dear Authors,

In this study, the authors focused on determining the effective contact time of seven disinfectants against eight bacterial strains from 30 s to 19 min. The results confirmed that Gram-negative bacteria were more resistant to the seven disinfectants than  Gram-positive bacteria. Moreover, the results showed that the type of surface to be disinfected does not affect the efficiency of the disinfectant used, especially on the types of surfaces tested in this study. The manuscript is very interesting, but some points should be addressed before the manuscript is considered for publication. So, the manuscript accepts after minor revision.

In the introduction section:

Line 7:  Please delete (and), repeated twice.

Line 49 & 50:  delete word (compound).

Line 21& 82: Correct the word ef-ficacy.

Line 52:  Correct (mul-tidrug-resistant) to (multidrugresistant).

In the Material and Method section:

Line 79:  delete (of)  in (to evaluate of the antibacterial effect).

Line 87:  Correct (multidrug resistant) to (multidrug -resistant).

Line 95:  change ( in classes phenolic compounds) to (classes of phenolic compounds).

Line 158:  correct (sur-face).

In the Discussion section:

Line 239:  Correct (multidrug resistant) to (multidrug -resistant.

Line 278:  add space between (at 3and 2min)

Line 293:  change (suggests a global increase in rates of resistance of Gram-negative bacteria [36-38]) to ( suggests a global increase in Gram-negative bacteria resistance rates [36-38]).

With my best wishes

Minor editing of English language required

Author Response

Response to Reviewer 3 Comments (Round 1)

Round 2

Reviewer 1 Report

Line 154: 2 ml -> 2mL

Line 140: cm 2 -> cm2

Author Response

Response to Reviewer 1 Comments (Round 2).

Reviewer 2 Report

I consider the manuscript suitable for publication.

Author Response

Thank you.